

# Validation of spectroscopic gas analyzer accuracy using gravimetric standard gas mixtures: Impact of background gas composition on CO$_2$ quantitation by cavity ring-down spectroscopy

Jeong Sik Lim, Miyeon Park, Jinbok Lee, Jeongsoon Lee

Center for Gas Analysis, Metrology for Quality of Life, Korea Research Institute of Standards and Science (KRISS), Gajeong-ro 267, Yuseong-gu, Daejeon 34113, Republic of Korea

*Correspondence to*: Jeongsoon Lee (leejs@kriss.re.kr)

**Abstract.** Effect of background gas composition on the measurement of CO$_2$ levels was investigated by wavelength-scanned

cavity ring-down spectrometry (WS-CRDS) employing a spectral line centered at the R(1) of the $(3\ 0^0\ 1)_{III} \leftarrow (0\ 0\ 0)$ band. For this purpose, eight cylinders with various gas compositions were gravimetrically and manometrically prepared within $2\sigma$ = 0.1 %, and these gas mixtures were introduced into the WS-CRDS analyzer calibrated against standards of ambient air composition. Depending on the gas composition, deviations between CRDS-determined and gravimetrically (or manometrically) assigned CO$_2$ concentrations ranged from -9.77 to 5.36 μmol/mol, e.g., excess N$_2$ exhibited a negative

deviation, whereas excess Ar showed a positive one. The total pressure broadening coefficients (TBPCs) obtained from the composition of N$_2$, O$_2$ and Ar thoroughly corrected the deviations up to -0.5–0.6 μmol/mol, while these values were -0.43–1.43 μmol/mol considering PBCs induced by only N$_2$. The use of TBPCs enhanced deviations to be corrected to ~0.15 %. Furthermore, the above correction linearly shifted CRDS responses for a wide extent of TPBCs ranging from 0.065 to 0.081 cm$^{-1}$ atm$^{-1}$. Thus, accurate measurements using optical intensity-based techniques such as WS-CRDS require TBPC-based

instrument calibration.

## 1 Introduction

Emission of carbon dioxide (CO$_2$), the most important greenhouse gas, has been reported to increase, resulting in global climate change (Messerschmidt et al., 2011; Solomon et al., 2007). According to the IPCC Fourth Assessment Report (Solomon et al., 2007), CO$_2$ is the major contributor to global warming, having a 62.9 % share of the total radiative force caused by long-lived greenhouse gases. Although it is not plausible to quantify its sources and sinks within considerably small uncertainties (Conway et al., 1988; Schulze et al., 2009), all countries have agreed to consistently control CO$_2$ emissions, necessitating

accurate measurements of atmospheric CO$_2$ mole fractions. Gas chromatography (GC) coupled with flame ionization detection (FID) (van der Laan et al., 2009), non-dispersive infrared spectroscopy (NDIR) at 4.26 μm (Lee et al., 2006; Min et al., 2009; Crawley, 2008; Tohjima et al., 2009), Fourier transform infrared (FTIR) spectroscopy (Griffith et al., 2012), tunable diode laser absorption spectroscopy (TDLAS) (Durry et al., 2010), wavelength-scanned cavity ring-down spectroscopy (WS-CRDS) (Crosson, 2008), and other cavity-enhanced absorption spectroscopies (O'Shea et al., 2013) are well-known techniques for

quantifying atmospheric CO$_2$. Despite exhibiting the advantage of high measurement precision, GC-FID suffers from long acquisition time due to delayed CO$_2$ retention in the separation column (typically a few tens of minutes). NDIR shows better performance than GC-FID in real-time measurements due to using filtered spectral fingerprints of CO$_2$ instead of relying on analyte separation. However, frequent calibrations are required to correct NDIR response drifts. Recently, WS-CRDS has attracted attention because of its high precision and low drift. In contrast to intensity-based techniques such as NDIR and





TDLAS, CRDS is immune to laser shot noise and detector electric noise due to employing the ring-down count method. Furthermore, the increased path length offered by the resonant optical cavity provides excellent sensitivity, i.e., signal-to-noise ratio, and high precision. Since a $CO_2$ inter-laboratory compatibility of ± 0.1 μmol/mol in the Northern Hemisphere was set as a goal by the World Meteorological Organization (WMO), WS-CRDS is viewed as a competitive technique for measuring

atmospheric greenhouse gas levels (Rella et al., 2013).

Accurate measurements of atmospheric $CO_2$ levels by WS-CRDS require the removal of water vapor, which causes spectral interference, and an empirical cubic polynomial model for correcting the water background has been developed (Rella et al., 2013). Nevertheless, $CO_2$ mole fraction measurements can be adversely affected by spectral line broadening if calibration gas mixtures whose background composition is different from the natural $N_2$:$O_2$:Ar ratio in the atmosphere are used (Nara et al.,

2012). In this study, standard gas mixtures containing ambient levels of $CO_2$ in synthetic air ($N_2 + O_2 + Ar$) were gravimetrically prepared for utilization as calibration standards and measuring targets for investigating the impact of background gas composition on WS-CRDS responses, owing to the excellent uncertainty of gravimetric gas mixtures. Furthermore, an empirical equation for correcting the "matrix effect" was derived in terms of total pressure broadening. The good agreement achieved between $CO_2$ mole fractions of the calibration standards and synthetic samples of arbitrary composition validated the

measurement accuracy of matrix-effect-corrected WS-CRDS.

## 2 Materials and methods

### 2.1 Preparation of standard gas mixtures

Gas mixtures were prepared using gravimetric and volumetric (or manometric) methods, based on ISO 6142 (International Standard, 2001) and ISO 6144 (International Standard, 2003), respectively. The gravimetric method featured filling pure $CO_2$

(MG industries, USA) and $N_2$ (Deokyang Energen, South Korea) gases into a clean aluminum cylinder. Subsequently, pure $O_2$ (Praxair Co., South Korea) and Ar (Deokyang Energen, South Korea) gases were added to the obtained $CO_2$/$N_2$ mixture to obtain an ambient level of $CO_2$ in a matrix of synthetic air. The amounts of filled gases were determined based on their weight, which was obtained by weighing the aluminum cylinder before and after filling. The weights used for calibrating the weighing balance (Mettler Toledo, XP 26003L, USA) were calibrated against the national kilogram standard to ensure measurement

traceability. For high weighing precision, an automatic weighing machine patented by KRISS was used to control the loading position on the weighing pan of the top loading balance, resulting in a typical weighing uncertainty of less than 0.005 %. A circular turntable was used to support tare and sample cylinders. During weighing, the drift of the weighing balance and the buoyancy effect exerted by the cylinders were effectively corrected or cancelled out by using the following bracketing sequence: tare – cylinder A – tare – cylinder B – tare – cylinder C. The preparation of standard gas mixtures based on this technique has

been reported in detail elsewhere (Wessel, 2008). The $CO_2$ mole fraction in the resulting mixture can be computed as follows:

$$y_j = \frac{\sum_{A=1}^{P}\left(\frac{x_{j,A}\cdot m_A}{\sum_{i=1}^{n} x_{i,A}\cdot M_i}\right)}{\sum_{A=1}^{P}\left(\frac{m_A}{\sum_{i=1}^{n} x_{i,A}\cdot M_i}\right)} \tag{1}$$

Here, $y_j$ is the mole fraction of component $j$ in the gas mixture, $P$ is the total number of parent gases, $n$ is the total number of components in the final mixture, $m_A$ is the measured mass of parent gas A, $M_i$ is the molar mass of component $i$, and $x_{i,A}$ or $x_{j,A}$ is the mole fraction of component $i$ or $j$ in parent gas A. Therefore, quantification of impurities present in pure parent gases is

needed to determine the composition of each parent gas. Hence, impurities in $N_2$, $O_2$, Ar, and $CO_2$ were analyzed by gas chromatography employing various detection methods, e.g., thermal conductivity detection (TCD), pulsed discharge detection (PDD), flame ionization detection (FID), and atomic emission detection (AED), with detector assignments for all impurities given in Table 1. Purity, namely the mole fraction of the dominant component in "pure" parent gas ($x_{pure}$) was determined as follows:



$$x_{\text{pure}} = 1 - \sum_{i=1}^{N} x_i \tag{2}$$

where $N$ is the number of impurities likely to be present in the final mixture. For selecting target impurities, the source and its purification process were considered. If the expected impurity was not detected, its mole fraction was set to half of the limit of detection (LOD/2), and the associated standard uncertainty was defined as the assigned mole fraction divided by $\sqrt{3}$, e.g.,

LOD/($2 \cdot \sqrt{3}$), as expected for a uniform probability density function ranging from 0 to LOD [International Standard, 2001]. In particular, it was very important to accurately analyze the mole fractions of target components ($N_2$, $O_2$, Ar, and $CO_2$) in the respective raw gases, since the weighed target component amount in the obtained mixture could be biased by the presence of the same component in other raw gases as an impurity. For instance, the mole fractions of $CO_2$ in pure $N_2$, $O_2$, and Ar gases were determined as 0.002, 0.195, and < 0.002 µmol/mol, respectively. Thus, the amounts of $CO_2$ in pure $N_2$ and Ar gases were

negligible and did not impact final mixtures with $CO_2$ fractions above 300 µmol/mol. However, the large amount of $CO_2$ in pure $O_2$ led to a bias of 0.04 µmol/mol, which was comparable to the uncertainty level of the final mixture. Table 1 summarizes the reference values and associated uncertainties of major impurities in raw gases.

For $CO_2$, a verification test was representatively performed to determine the potential systematic error of the gravimetric procedure described above, relying on comparing the detection sensitivity of $CO_2$ in different gas mixtures using GC-FID

coupled with an MS-5A (molecular sieve 5A, 4 m) separation column. The column oven was kept at 30 °C, and ultra-high-purity nitrogen (99.999 %, Deokyang Energen) was used as a carrier gas. Sample gas flows were carefully controlled to ensure that the same amount of gas was introduced into the sample loop regardless of its composition; for this purpose, mass flow controllers (MFCs) were calibrated using a flow meter (Digital flow calibrator (cat#20123), Restek Inc., USA). Therefore, the $CO_2$ mole fraction uncertainty of prepared mixtures included uncertainties associated with the weighing process, raw gases

purities, and verification tests, resulting in a gravimetric preparation uncertainty of less than 0.1 µmol/mol ($1\sigma$). The standard gas mixture denoted as EBXXXXXXX (Table 2) was prepared by the static volumetric method (International Standard, 2003; Waldén, 2009). The mole fraction of $N_2$ was varied by diluting dry air with high-purity $N_2$ (> 99.999 %), and the mixing ratio was estimated using the measured pressure ratio of filled gases. $CO_2$ mole fractions of three manometric cylinders (EBXXXXXXX) were finally confirmed by comparison against gravimetric standards (Table 2). Notably, the prepared gas

mixtures were maintained under very dry conditions, with the mole fraction of $H_2O$ being less than 5 µmol/mol.

## 2.2 Cavity ring-down spectroscopy

Cavity ring-down spectroscopy (CRDS) as an ultrasensitive technique introduced by O'Keefe and Deacon in 1988 (Chen et al., 2010; Rothman et al., 2005). In principle, the leakage rate of the trapped laser source in the optical cavity can be fitted by monoexponential decay, and absorbance at wavelength $\lambda$ can then be calculated from the difference of ring-down signal decay

rates in the presence and absence of the target gas. Alternatively, the absorbance at $\lambda$ can be determined from the ring-down time at the non-absorbing wavelength $\lambda_0$ in the presence of the target gas. In this study, a commercial wavelength-scanned cavity ring-down spectrometer (WS-CRDS, G-1301, Picarro, USA) was employed. Since the WS-CRDS system has been described elsewhere (Chen et al., 2010; Nara et al., 2012), only a brief description is provided here. The WS-CRDS analyzer, operating at a wavelength of 1.603 µm that corresponds to R(1) of the $(3\,0^0\,1)_{\text{III}} \leftarrow (0\,0\,0)$ band, comprised diode lasers, a

high-precision wavelength monitor, a high-finesse cavity defined by three high-reflectivity mirrors (<99.995 %), a photodiode detector, and a data acquisition computer. Laser light confined in the cavity traveled along the triangular optical axis, exhibiting an effective path length of 15–20 km. Ambient air or gas from a pressure-regulated tank was supplied to the optical cavity through a built-in diaphragm pump, which was conditioned to a highly controlled pressure and temperature of $140 \pm 0.05$ Torr and $40 \pm 0.01$ °C, respectively.

For this study, a gas flow rate of 400 mL/min and a pig-tailed bypass-out were combined to achieve a steady gas flow undisturbed by laboratory pressure fluctuation, yielding a constant pressure in the CRDS cavity (Fig. 1). The inner diameters



of stainless steel tubes connecting highly pressurized cylinders to the MFC (5850E, Brooks Inc., USA) inlet and the MFC to the spectrometer equaled 1/8 and 1/16 inch, respectively. High-purity nitrogen was used for flushing the gas lines and CRDS analyzer between switching cylinders.

The measured spectral line consisting of ~10 points was fitted by the Galatry profile to obtain quantitative information, based
on the assumption that the CRDS read-out was influenced only by variations in the $CO_2$ concentration of tested samples, and not by variations of background gas composition (Chen et al., 2010). This assumption implies that the peak height of the fitted profile was regarded as a CRDS read-out instead of the corresponding integrated area (Nara et al., 2012). CRDS responses were calibrated against gravimetric standards with $CO_2$ concentrations very similar to those of ambient air (between 360 and 410 µmol/mol). Absorbance was found to be linearly proportional to the concentration of light-absorbing gas, as indicated by
the straight-line fit of CRDS responses with $R^2 \sim 0.9999$ (Fig. 2 and Table 3), supporting the validity of the attempted calibration and the hypothesis proposed in this study. In other words, deviations from expected sensitivity (i.e., CRDS response divided by the gravimetric concentration of $CO_2$) were due to deviations in the composition of background gas from that of ambient air, namely the extent of alien gas line broadening or narrowing.

## 3 Results & discussion

To investigate the effect of background gas composition on CRDS responses, gas mixtures were analyzed against ambient-air-like standards using a well-calibrated CRD spectrometer (Table 4).

Deviations of $CO_2$ concentrations determined by CRDS from those assigned by gravimetry (or manometry) ranged from −2.44 to 1.39 %. CRDS responses of EB0006391 and ME0434 were in good agreement with the assigned $CO_2$ concentrations,
showing deviations of less than 0.1 µmol/mol, whereas extreme deviations of greater than 1 % were observed for cylinders DF4560 and ME5537. In particular, the $CO_2$ concentration of DF4560 ($CO_2$ in pure $N_2$) showed a deviation of −9.77 µmol/mol. Therefore, it can be conjectured that $N_2$-induced broadening is more important than that induced by other background gases, $O_2$ and Ar. Since the optical cavity was kept at constant pressure and temperature, Doppler broadening was not considered. Instead, collision-induced broadening (or narrowing) was invoked in the case of variable composition. The collisional half-
width, i.e., the total pressure broadening coefficient ($\gamma_{TPB}$), can be expressed as follows:

$$\gamma_{TPB} = \sum_{i=1}^{n} \gamma_i \cdot p_i \tag{3}$$

where $\gamma_i$ is the pressure broadening coefficient (PBC) of component $i$, and $p_i$ is the partial pressure of component $i$, e.g., its molar fraction multiplied by the cavity pressure of 18 kPa. The maximum peak height of the Galatry profile at a given background gas composition, $G(\gamma)$, can be assumed to be linearly proportional to the PBC for a sufficiently narrow interval of
$p_i$, $\Delta p_i$ (Varghese and Hanson, 1984). In view of the dominance of $N_2$-induced pressure broadening, the difference between CRDS-determined and assigned $CO_2$ concentrations of the measured sample, $D_{STD\text{-}CRDS}$, can be determined as follows:

$$D_{STD-CRDS} \propto G(\gamma) \propto \gamma_{N_2} \cdot p_{N_2} \tag{4}$$

As shown in Fig. 3, a linear relationship between $D_{STD\text{-}CRDS}$ and $N_2$-induced line broadening was found at given partial pressures (i.e., mole fractions multiplied by cavity pressure) in the optical cavity.

The PBC of $N_2$ was set to 0.08064 $cm^{-1}$ $atm^{-1}$, as reported by Nakamich et al. (2006). Since $N_2$ showed the largest PBC among those of other background components, positive (or negative) deviations between CRDS-determined and assigned $CO_2$ concentrations of tested cylinders, i.e., the lower (or higher) extent of pressure broadening, were observed at $N_2$ concentrations below (or above) the ambient value of 78 cmol/mol corresponding to ME5590 (Table 4). Thus, the $CO_2$ concentration could be corrected based on the following linear fit:



$$y_{corrected} = y_{CRDS} - \left(-606.63 \cdot \gamma_{N_2} \cdot p_{N_2} + 38.656\right) \tag{5}$$

where $\gamma_{N_2} \cdot p_{N_2}$ is the $N_2$-induced pressure broadening, $y_{CRDS}$ is the value obtained by WS-CRDS, and $y_{corrected}$ is the $CO_2$ concentration corrected for $N_2$-induced pressure broadening. Corrected $CO_2$ concentrations exhibited good agreement (within 0.4 %) with the regression fit ($R^2 \sim 0.9736$). This correction error significantly exceeded the instrumental precision (reported as 0.01 % ($1\sigma$); Nara et al., 2012), strongly suggesting the presence of other error sources.

The pressure broadening correction of ME5537 showed the highest deviation of 0.4 %. The background gas composition of ME5537 (70.98 % $N_2$, 18.85 % $O_2$, and 10.13 % Ar) implied that the Ar content should be taken into account for the correction. Since $CO_2$ self-broadening is negligible due to the low concentration of $CO_2$ compared to that of other components ($N_2$, $O_2$, and Ar) in the investigated gas mixtures, the total pressure broadening coefficient (TPBC) could be expressed as a function of alien gas PBCs and the partial pressures of the corresponding components:

$$\gamma_{TPBC} = \gamma_{N_2}p_{N_2} + \gamma_{O_2}p_{O_2} + \gamma_{Ar}p_{Ar} \tag{6}$$

Table 5 shows the reported PBCs for $N_2$, $O_2$, and Ar, and Table 6 shows TPBCs of all cylinders, with (a), (b), and (c) denoting results obtained independently by Pouchet et al. (2004), Nakamichi et al. (2006), and HITRAN2004, respectively.

Since the coefficients of Ar have not been reported by Pouchet et al. (2004) and HITRAN2004, the corresponding TPBCs include only $N_2$- and $O_2$-related pressure broadening (Table 6). Therefore, the TPBCs in (a) and (c) were underestimated in comparison to that in (b). For instance, TPBCs of 0.0636 and 0.0685 were obtained for cylinder ME5537 in the cases of (a) and (c), respectively, with the value for (b) equaling 0.07625. As shown in Table 6, the TPBC of ME5537 exhibited the largest deviation of 20 %, originating mainly from the Ar mole fraction. Figure 4 shows $D_{STD\text{-}CRDS}$ values (column 4 of Table 7: $(B - A)^C$) as a function of calculated TPBCs (taken from Table 6).

TPBC values reported by Nakamichi et al. (2006) exhibited a linear correlation with CRDS responses within the investigated background composition interval. In practice, Huang and Yung (2004) reported that the Lorentzian width is inversely proportional to the peak value of the Voigt function for a fixed Gaussian width. The results shown in Fig. 4 reveal that $D_{STD\text{-}CRDS}$ values decreased with increasing TPBCs, in agreement with previous reports (Huang and Yung, 2004). Only the result of (b) exhibited a fairly linear behavior; however, non-linearity was observed when the broadening coefficients of $O_2$ or Ar were not taken into account. The following equation was derived for correcting CRDS-determined concentrations:

$$y_{corr.TPB} = y_{CRDS} - \left(-3382.1 \cdot \gamma_{TPBC} + 262.65\right) \tag{7}$$

Here, $y_{CRDS}$ is the CRDS-measured value of the standard gas mixture, and $y_{corr.TPB}$ is the corresponding corrected CRDS response computed using the relation in (b) (Fig. 4). Table 7 summarizes the results obtained after correction using Eq. (7), showing that the correction was improved from 0.68 ($N_2$ PBC) to 0.33 µmol/mol (TPBC) in terms of standard deviations ($1\sigma$) of differences (corrected minus gravimetry-assigned). Furthermore, $R^2$ was improved to 0.99 when pressure broadening related to three main components of air ($N_2$, $O_2$, and Ar) was taken into account. For every cylinder, excellent agreement was observed after implementing the TPBC corresponding to the assigned values. In particular, even cylinders DF4560, ME5590, and ME5537, whose background gas compositions were significantly different from that of ambient air, exhibited good correlation of $CO_2$ concentrations determined by CRDS with those assigned by gravimetry or manometry.

## 4 Conclusions

In this study, we investigated the impact of background gas composition on spectroscopic quantitation of $CO_2$ at ambient concentration. Standard gas mixtures with various background compositions were prepared by gravimetry or manometry for





use as calibration standards and test samples. Purity analysis and gravimetric weighing showed high accuracy and precision. For purity analysis, analytical techniques such as GC-PDD, TCD, FID, AED, and dew point metering were used. Raw gas ($N_2$, $O_2$, Ar, and $CO_2$) purities were obtained within uncertainties of less than 0.001 % ($1\sigma$). Moreover, biasing impurities in $N_2$, $O_2$, and $CO_2$ were accurately crosschecked. With a weighing precision of 0.007 %, the preparation uncertainties of gravimetric

and volumetric mixing were demonstrated to be lower than 0.05 and 0.1 % ($2\sigma$), respectively, after performing verification tests. The preparation uncertainty of manometry was slightly higher than that of gravimetry, still being sufficiently satisfactory to distinguish error sources for "matrix effect" correction. Based on the composition accuracy of the prepared gas mixtures, $CO_2$ levels were determined by WS-CRDS for eight standard gas mixtures with different background compositions. An injection unit with a bypass-out was used to ensure a precise and moderate gas inflow from a highly pressurized cylinder to

the WS-CRD spectrometer, which was calibrated against well-certified standard gas mixtures of air composition with $CO_2$ levels of 360–410 $\mu mol/mol$. Among the eight cylinders, the CRDS responses of EB0006391 and ME0434 were well-matched to the corresponding preparative values, whereas the values obtained for other cylinders exhibited large deviations between +5.36 and −9.77 $\mu mol/mol$. For a $N_2$-enriched mixture (DF4560), the CRDS-determined $CO_2$ concentration was 2.44 % lower than the preparative value. Since CRDS calibration was performed using standards with ambient air composition, the fact that

CRDS responses tended to be negative for $N_2$-enriched and positive for Ar-enriched mixtures was in good agreement with the results obtained in earlier experimental (Nara et al., 2012; Zhao et al., 1997) and theoretical studies (Huang and Yung, 2004), reflecting the dependence of line broadening on alien gas composition.

Therefore, a linear shift of CRDS responses was observed for TPBCs above 0.05 $cm^{-1}$ $atm^{-1}$, which covers 20 % $N_2$-enriched and 10 % Ar-enriched gas mixtures. TPBC-corrected CRDS responses were in good agreement with the gravimetric (or

manometric) concentration of the investigated gas mixtures within 0.15 % (± 0.6 $\mu mol/mol$). Considering the instrumental uncertainty of 0.01 % ($1\sigma$), the improved PBC uncertainties should lead to lower discrepancies of corrected CRDS responses. The correction presented in Eq. (7) works only for the designated vibrational transition, i.e., R(1) of the $(3\ 0^0\ 1)_{III} \leftarrow (0\ 0\ 0)$ band at 1.603 $\mu m$, and referred PBCs, but a similar calibration strategy can be used for determining gas mixing ratios by other intensity-based optical measurement techniques.

**Code availability:** Not applicable

**Data availability:** Not applicable

**Appendices:** None

**Supplement link:** N/A

**Team list:** Jeong Sik Lim, MiYeon Park, Jinbok Lee, and Jeongsoon Lee

**Author contribution:** Jinbok Lee prepared the certified reference materials, J. Lim and M. Park performed measurements and analysis. Jeongsoon Lee designed experiments, and J. Lim prepared the manuscript with contributions from other co-authors.

**Competing interests:** The authors declare that they have no conflict of interest.

**Disclaimer:** None.

**Acknowledgements:** This work was funded by the Korea Meteorological Administration Research and Development Program
under Grant No. KMIPA 2015-2032. No data sharing issues exist, as all numerical information is provided in the figures produced by solving the equations in this paper. All data in this paper are properly cited and referred to in the reference list.



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





**Table 1.** Purities of raw carbon dioxide and background gases ($N_2$, $O_2$, and Ar).

| Impurity Component | Mole fraction [µmol/mol] | | | | Detectors[1] |
|---|---|---|---|---|---|
| | $CO_2$ | $N_2$ | $O_2$ | Ar | |
| $H_2$ | <0.1 | <0.1 | <0.1 | <0.1 | PDD[2] |
| $O_2$ | <0.1 | 0.003 ± 0.003 | - | 0.003 ± 0.002 | PDD |
| Ar | <0.1 | 21.6 ± 4.32 | <1.0 | - | TCD[3] |
| $N_2$ | 12.8 ± 2.56 | - | 3.1 ± 0.62 | 2.4 ± 0.48 | PDD |
| CO | 0.3 ± 0.06 | <0.005 | 0.08 ± 0.016 | <0.005 | PDD and FID[4] |
| $CH_4$ | 2.6 ± 0.52 | <0.005 | <0.005 | <0.005 | PDD and FID |
| $CO_2$ | - | 0.002 ± 0.001 | 0.195 ± 0.039 | <0.002 | PDD and FID |
| $H_2O$ | 4.5 ± 2.25 | 1.6 ± 0.8 | 1.1 ± 0.55 | 0.9 ± 0.45 | Dew point meter |
| $C_2$ | 2.8 ± 0.56 | - | - | - | AED[5] |
| $C_3–C_5$ | 0.7 ± 0.35 | - | - | - | AED |
| Purity (%) ($k = 2$) | 99.9976± 0.0007 | 99.9976 ± 0.0009 | 99.9995 ± 0.0002 | 99.9996 ± 0.0001 | |

1. Tabulated detectors were coupled to the main body of the gas chromatograph (Agilent 6890A)

2. Pulsed discharge detector

3. Thermal conductivity detector

4. Flame ionization detector

5. Atomic emission detector





**Table 2.** Mole fractions of gas mixtures.

| Cylinder # | Gas composition [cmol/mol] | | | | Preparation method |
|---|---|---|---|---|---|
| | $CO_2$[1] | $N_2$ | $O_2$ | Ar | |
| DF4560 | 400.61 (0.05%) | 99.96 | - | - | gravimetry |
| EB0011591 | 351.78 (0.10%) | 83.45 | 16.48 | 0.04 | manometry |
| EB0011528 | 353.08 (0.10%) | 80.97 | 18.19 | 0.81 | manometry |
| ME5590 | 386.94 (0.05%) | 78.33 | 21.63 | - | gravimetry |
| EB0006391 | 406.40 (0.10%) | 78.16 | 20.87 | 0.93 | manometry |
| ME0434 | 402.25 (0.05%) | 78.07 | 21.03 | 0.87 | gravimetry |
| ME5502 | 384.35 (0.05%) | 77.57 | 20.53 | 1.86 | gravimetry |
| ME5537 | 385.35 (0.05%) | 70.98 | 18.85 | 10.12 | gravimetry |

1. Numbers denote the mole fraction (µmol/mol) of $CO_2$ and its relative
preparation uncertainty





**Table 3.** Summary of CRDS calibration results.

| Cylinder # | CO$_2$ mole fraction [μmol/mol] | | | Difference | |
|---|---|---|---|---|---|
| | Gravimetrically assigned value (A) | Before CRDS calibration | After CRDS calibration (B) | (B − A) [μmol/mol] | (B − A) / A × 100 [%] |
| ME0424 | 371.22 | 371.18 | 371.29 | 0.07 | 0.0193 |
| ME0485 | 380.31 | 380.23 | 380.28 | −0.03 | −0.0088 |
| ME5552 | 384.76 | 384.66 | 384.67 | −0.09 | −0.0222 |
| ME0434 | 402.25 | 402.41 | 402.30 | 0.05 | 0.0117 |





**Table 4.** $CO_2$ concentrations determined by gravimetry and measured by well-calibrated CRDS, together with the correction due to $N_2$-induced pressure broadening. Differences between the measured (corrected) and assigned concentrations are also listed.

| Cylinder # | $CO_2$ mole fraction [µmol/mol] | | | Difference | | |
|---|---|---|---|---|---|---|
| | Gravimetrically assigned value (A) | CRDS measured value (B) | PBC ($N_2$) corrected (C) | (B − A) [µmol/mol] | (B − A) / A × 100 [%] | (C − A) / A × 100 [%] |
| DF4560 | 400.61 | 390.84 | 401.09 | −9.77 | −2.44 | 0.12 |
| EB0011591 | 351.78 | 349.62 | 351.79 | −2.16 | −0.61 | 0.00 |
| EB0011528 | 353.08 | 352.05 | 353.00 | −1.03 | −0.29 | −0.02 |
| ME5590 | 386.94 | 386.51 | 386.17 | −0.43 | −0.11 | −0.20 |
| EB0006391 | 406.40 | 406.39 | 405.97 | −0.01 | 0.00 | −0.11 |
| ME0434 | 402.25 | 402.34 | 401.87 | 0.09 | 0.02 | −0.09 |
| ME5502 | 384.35 | 384.80 | 384.09 | 0.45 | 0.12 | −0.07 |
| ME5537 | 385.35 | 390.71 | 386.78 | 5.36 | 1.39 | 0.37 |




**Table 5.** Summary of $N_2$-, $O_2$-, and Ar-related pressure broadening coefficients in $cm^{-1}$ $atm^{-1}$.

|  | Pouchet et al. | Nakamichi et al. | HITRAN |
| --- | --- | --- | --- |
| $\gamma_{N_2}$ | 0.0721 | 0.08064 | 0.0778 |
| $\gamma_{O_2}$ | 0.0660 | 0.06695 | 0.0702 |
| $\gamma_{Ar}$ | - | 0.06312 | - |
| $\gamma_{air}$ | - | - | 0.0758 |



**Table 6.** Pressure broadening for investigated gas mixtures based on pressure broadening coefficients from different sources.

| Cylinder # | Pouchet et al.[1] | Nakamichi et al.[2] | HITRAN |
|------------|-------------------|---------------------|--------|
| DF4560 | 0.0721 | 0.08061 | 0.0778 |
| EB0011591 | 0.0710 | 0.07835 | 0.0765 |
| EB0011528 | 0.0704 | 0.07798 | 0.0758 |
| ME5590 | 0.0708 | 0.07765 | 0.0761 |
| EB0006391 | 0.0701 | 0.07759 | 0.0755 |
| ME0434 | 0.0702 | 0.07758 | 0.0755 |
| ME5502 | 0.0695 | 0.07747 | 0.0748 |
| ME5537 | 0.0636 | 0.07625 | 0.0685 |

1 and 2 denote values obtained in each study.





**Table 7.** Comparison between assigned and TPBC-corrected $CO_2$ concentrations of investigated gas mixtures.

| Cylinder # | $CO_2$ mole fraction [µmol/mol] | | Difference | |
| --- | --- | --- | --- | --- |
| | Assigned value | TPBC-corrected value | (B − A) | (B − A) / A × 100 |
| | (A) | (B) | [µmol/mol] | [%] |
| DF4560 | 400.61 | 400.82 | 0.21 | 0.05 |
| EB0011591 | 351.78 | 351.97 | 0.19 | 0.05 |
| EB0011528 | 353.08 | 353.15 | 0.07 | 0.02 |
| ME5590 | 386.94 | 386.47 | −0.47 | −0.12 |
| EB0006391 | 406.40 | 406.15 | −0.25 | −0.06 |
| ME0434 | 402.25 | 402.09 | −0.16 | −0.04 |
| ME5502 | 384.35 | 384.17 | −0.18 | −0.05 |
| ME5537 | 385.35 | 385.95 | 0.60 | 0.16 |





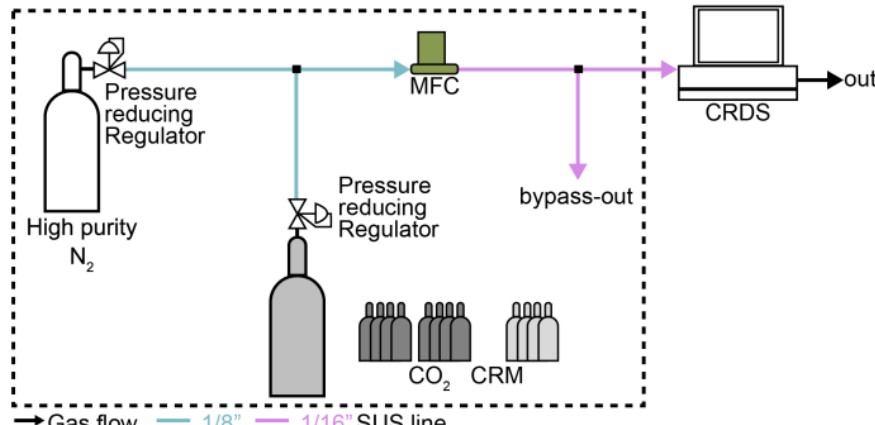

Figure 1: Schematic diagram depicting the gas supply to the WS-CRDS analyzer. The acronym SUS represents the stainless steel.




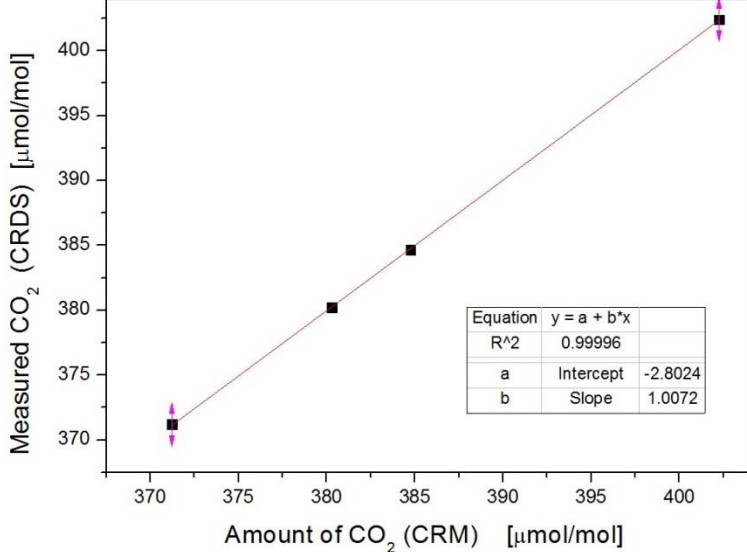

Figure 2: Result of WS-CRDS calibration using gravimetric standards (ambient air background composition, see main text for details). Good agreement between gravimetric and CRDS-determined $CO_2$ concentrations was observed.





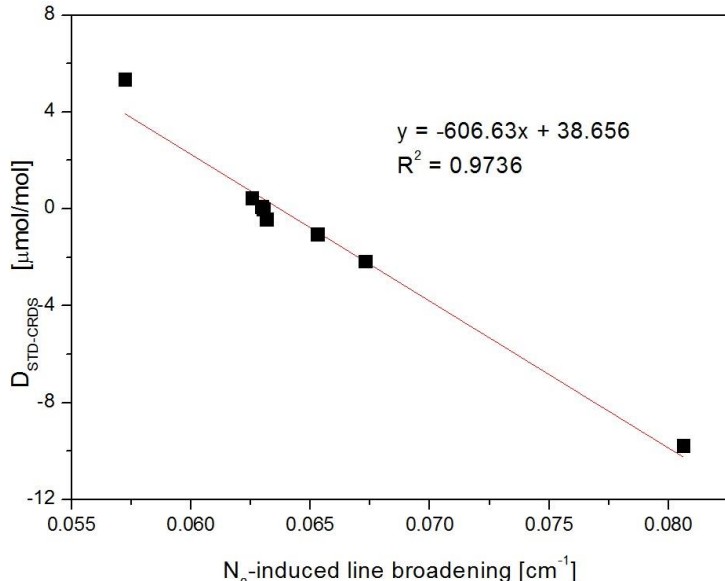

Figure 3: N$_2$-induced line broadening (*x*-axis) vs. difference between CRDS-measured and assigned CO$_2$ levels of standard gas mixtures (*y*-axis).





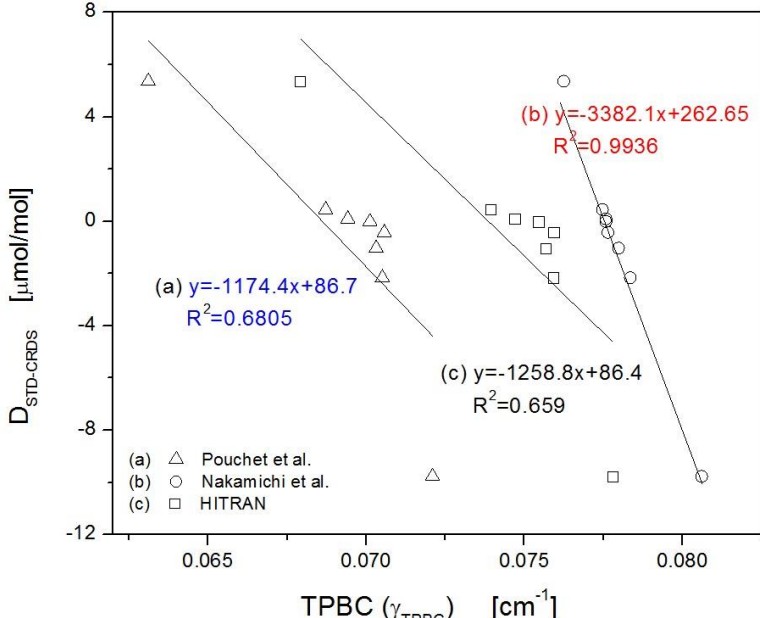

Figure 4: Total pressure broadening coefficient vs. difference between CRDS-measured and assigned $CO_2$ levels of standard

gas mixtures. Due to the lack of $\gamma_{Ar}$, correlations (a) and (c) exhibit poor fits.