# Peer review of "Validation of spectroscopic gas analyzer accuracy using gravimetric standard gas mixtures: Impact of background gas composition on CO$_2$ quantitation by cavity ring-down spectroscopy"

_Atmospheric Measurement Techniques, 2017_

## Referee Comment (RC1) · J. Kim (Referee) · 20 Jun 2017

**General Comments**

As the authors state, the CRDS technique is quickly becoming the preferred method for greenhouse gas measurements. While the improved precisions and stability in these instrument have helped made them popular, some of the short-comings associated with this technique, such as the potential for pressure broadening effects discussed here, are often overlooked. The experiment presented here showcase the metrological and instrumental expertise of the authors in addressing the pressure broadening effect from matrix gas composition, and overall the study is thoroughly done, certainly worthy of publication in my view.

I have one general comment. I note that two tanks that had close-to-ambient ratios of $N_2/O_2/Ar$, namely EB0006391 and ME0434, showed excellent agreement with values derived from CRDS prior to any correction (-0.01 and 0.09 umol/mol, respectively, in Table 4), and the TPBC corrected values actually get worse. In addition, while the TPBC corrections overall seem to make a positive impact, the correction errors still remain quite larger than the 0.01% instrument precision error that the authors suggest should be the ultimate goal. Do the authors have any comments on what other error sources could remain that would explain these results (some of which seems to already be present at the end of the discussion section)?

And one general question: Can the authors think of any scenarios outside of creating standard tanks from scratch that the TPBC correction would be necessary or beneficial?

**Specific Comments**

As the authors will know, the WS-CRDS technique measures only the main $^{12}CO_2$ isotopologue. I think any effect from this can effectively be canceled out if all of the gas mixtures used in this study (listed in both Table 2 and 3) used $CO_2$ from the same source cylinder. I wonder if this is indeed the case, and whether the authors should briefly address this point somewhere in the manuscript.

P3-L13: Was any correction to the concentrations applied based on the verification test on the GC, and if so how much? The authors state the verification test results were excellent (0.05 and 0.1 % $2\sigma$), but it would be interesting to see if those that looked worse in the verification test also showed larger deviation in the TPBC corrections. Perhaps this could be added in a supplementary section?

P3-L20: I think a more detailed description is needed for the static volumetric standard gas section. For example, line 22 mentions "dry air", is this some $CO_2$-free zero air that was used as the "complementary gas" (using the terminology in ISO 6144), or does it just refer to what was already in the tank prior to the "high-purity $N_2$" injection? Line 24 says the concentrations of the manometric cylinders were "confirmed" against the gravimetric standards: I would like clarification on whether the independent manometric values were confirmed by measurements against the gravimetric standards (on GC-FID?), and if so how the manometric vs gravimetric values compared, or if the values in the manometric tanks were "determined" from measurements against the gravimetric tanks. If the values were only confirmed, it would be nice to see how the values compared, perhaps in a supplementary section.

P4-L18: The numbers for the y-scale shown in Figure 4 (roughly -10 ~ 5.5?) do not seem to match those in column 4 of Table 7 (-0.47 ~ 0.60), but instead those in Table 4. Authors should check that this is only a graphing error and do not affect the conclusions of the paper.

Tables 4 and 7: I understand the logic of the authors' choice of separating the two tables to match the flow of the manuscript, however I do find myself frequently comparing the the N2-only vs TPBC corrected results. As such I would suggest that they be combined into one table, to represent an overview of the findings reported in this work, but I will leave that for the authors to decide.

**Technical Corrections**

P1-L29: "not plausible" suggests that this can't be done in the future, which may be true, but we should still remain hopeful that substantial progress in the modeling front can still be made. Perhaps change to "not yet feasible" instead?

P3-L20: I would suggest that the authors start a new paragraph for the section on the volumetrically prepared tanks.

P3-L22: Is the "high-purity $N_2$" used in the dilution different from the "ultra-high-purity nitrogen" mentioned in line 15? If they are the same, then I would advise using the same naming scheme for both.

P3-L24: "comprised" -> "is comprised of"

P3-L25: Perhaps mention which of the tanks reflect ratios close to ambient? I assume EB0006391 and ME0434?

P3-L40: "through a built-in diaphragm pump": Technically, I believe the pump pulls a vacuum after the cavity cell, whereas the authors' description gives the impression that air may go through the diaphragm pump into the cavity cell. Suggest editing this sentence to avoid ambiguity.

P3-L41: "inner" -> Did the authors mean "outer"?

P4-L8: "gravimetric standards" -> add "described in Table 3" after. How were these standards prepared in terms of $N_2$, $O_2$, and Ar? I assume at ambient ratios? This may be an important point, as the authors use the calibrations from these tanks as "truth".

p5-L13: Include reference for "HITRAN2004"?

P5-L16: "that" -> "those"

Table 6: I do not follow the author's foot note "1 and 2 denote values obtained in each study" for this table. I assume the numbers in this table were derived using the PBC's in Table 5 with the known $N_2$, $O_2$, and Ar ratios? But, aren't the HITRAN numbers calculated the same way, or am I mistaken? The footnote almost seems more appropriate for Table 5, where the PBC values in the table were taken from each study, but then are the HITRAN numbers different in this regard? Please clarify.

---

## Referee Comment (RC2) · Z. M. Loh (Referee) · 15 Jul 2017

This is a nice, succinct paper characterizing the effect of the background gas composition on CO2 measurement via cavity ringdown spectroscopy. This work should be useful to the atmospheric trace gas community as the number of atmospheric CO2 measurements (particularly via cavity ringdown spectroscopy (CRDS)) are likely to rise in the near term, potentially being made by non-experts. This paper is accessible and

clear and highlights a trap that many users of CRDS may fall into, by using industrially prepared calibration standards.

General Comments:

The authors present a set of total pressure broadening coefficients (TPBCs) that substantially improve agreement between CRDS determined $CO_2$ mixing ratios and the mixing ratios assigned to each tank during gravimetric or manometric preparation. However, the use of TPBCs does not reduce the discrepancy to within the World Meteorological Organization's $CO_2$ inter-laboratory compatibility goal of +/- 0.1 umol/mol (in the Northern Hemisphere, and 0.05 umol/mol in the Southern Hemisphere). As such, I would urge the authors to consider appending something similar to the following to the end of their abstract.

P1, L20: "... instrument calibration, or better still, use standards prepared with ambient air."

Additionally, I would like the authors to consider adding a sentence or two to this effect in their discussion section.

A further comment is that the authors do not mention the isotopic composition of the $CO_2$ used to prepare their synthetic standards. While I assume all eight standards were prepared with the same batch of $CO_2$ (and thus having the same $CO_2$ isotopic composition), this is worth mentioning (and handling) explicitly (preferably with the delta13C-$CO_2$ of the pure $CO_2$ used). As CRDS is a single line spectroscopic technique, it is inherently isotopologue specific. Therefore, using a pure $CO_2$ source with a significantly different isotopic composition from the background atmosphere will induce a systematic bias in CRDS determinations of mixing ratio unless this effect is accounted for. The authors already cite Lee et al. (2006), which deals with this question (though for NDIR rather than CRDS (for which the problem is at its most extreme)), so I assume they are familiar with the issue.

Specific technical comments:

P1 L28, consider inserting 'all' between quantify and its, and remove "considerably".

P3 L20, gases to become 'gas'

────────────────────────

---

## Author Comment (AC1) · 20 Jul 2017

The authors appreciate Dr. Loh's kind consideration of this manuscript. Please find our replies to the referee comments below.

General Comments

1. The authors present a set of total pressure broadening coefficients (TPBCs) that

substantially improve agreement between CRDS determined CO2 mixing ratios and the mixing ratios assigned to each tank during gravimetric or manometric preparation. However, the use of TPBCs does not reduce the discrepancy to within the World Meteorological Organization's CO2 inter-laboratory compatibility goal of +/- 0.1 umol/mol (in the Northern Hemisphere, and 0.05 umol/mol in the Southern Hemisphere). As such, I would urge the authors to consider appending something similar to the following to the end of their abstract.

P1, L20: "... instrument calibration, or better still, use standards prepared with ambient air."

Additionally, I would like the authors to consider adding a sentence or two to this effect in their discussion section.

- Thank you for the suggestion. Authors will add sentence as follow.

- P1, L20: ". . .. Instrument calibration or use standards prepared in same background composition of ambient air.

- The authors conjecture that major error sources arose from the mole fraction uncertainties of major components, e.g. N2, O2, Ar and CO2, and uncertainty of pressure broadening coefficients. According to this opinion, the authors will add sentences at the end of discussion section as follow.

- "It is worth noting that the quality of the TPBC correction can be improved further by using quality standards with lower composition uncertainties, including 13CO2 isotopologues and precisely measured broadening coefficients that are deduced from advanced line-shape functions such as Galatry and Rautian profiles."

- With regard to the isotopes ratio, please see the reply for general comment 2.

2. A further comment is that the authors do not mention the isotopic composition of the CO2 used to prepare their synthetic standards. While I assume all eight standards were prepared with the same batch of CO2 (and thus having the same CO2 isotopic composition), this is worth mentioning (and handling) explicitly (preferably with the $\delta$13CCO2 of the pure CO2 used). As CRDS is a single line spectroscopic technique, it is inherently isotopologue specific. Therefore, using a pure CO2 source with a significantly different isotopic composition from the background atmosphere will induce a systematic bias in CRDS determinations of mixing ratio unless this effect is accounted for. The authors already cite Lee et al. (2006), which deals with this question (though for NDIR rather than CRDS (for which the problem is at its most extreme)), so I assume they are familiar with the issue.

- The authors understand this comment is very similar to first specific comment of RC1. The 12/13 ratio of CO2 raw gas for gravimetric standards was similar to the atmospheric level approximately -11‰ The volumetric standards with prepared with the dry air and high purity N2 (>99.999%). This suggests similar isotope ratios would occur across the prepared cylinders. For verification (calibration) of prepared gravimetric (volumetric) standards, the CO2 mole fractions in them were verified by GC-FID, which measured total carbon isotopes. Therefore, the isotope effect were hardly discernable in this study. However, it might be the case that the isotope ratios of CO2 in the "dry air" can vary or deviate from the CO2 raw gas to cause some extent of discrepancy in the CRDS response. The authors will add sentences at the end of the section 2.1 as follow.

- The 12C/13C ratio of CO2 raw gas for the gravimetric standards was similar to the atmospheric level of approximately -11‰ which suggests similar isotope ratios would occur across the prepared cylinders as determined by gravimetry and volumetry. Nevertheless, isotope effects biasing the CRDS response seemed to be hardly discernable in this study because verification (calibration) of the CO2 mole fractions in the prepared gravimetric (volumetric) standards was carried out by GC-FID, which measured the total carbon isotopes."

Specific Comments

1. P1 L28, consider inserting 'all' between quantify and its, and remove "considerably"

- It will be corrected as suggested.

2. P3 L20, gases to become 'gas'

- It will be corrected as pointed out

Please also note the supplement to this comment:
https://www.atmos-meas-tech-discuss.net/amt-2017-54/amt-2017-54-AC1-supplement.pdf

**Supplement:**

**Reply to RC2**

Jeongsoon Lee (Corresponding author)
leejs@kriss.re.kr

The authors appreciate Dr. Loh's kind consideration of this manuscript. Please find our replies to the referee comments below.

**General Comments**

1.  The authors present a set of total pressure broadening coefficients (TPBCs) that substantially improve agreement between CRDS determined $CO_2$ mixing ratios and the mixing ratios assigned to each tank during gravimetric or manometric preparation. However, the use of TPBCs does not reduce the discrepancy to within the World Meteorological Organization's $CO_2$ inter-laboratory compatibility goal of +/- 0.1 umol/mol (in the Northern Hemisphere, and 0.05 umol/mol in the Southern Hemisphere). As such, I would urge the authors to consider appending something similar to the following to the end of their abstract.

    P1, L20: "... instrument calibration, or better still, use standards prepared with ambient air."

    Additionally, I would like the authors to consider adding a sentence or two to this effect in their discussion section.

    -   Thank you for the suggestion. Authors will add sentence as follow.
    -   P1, L20: ".... Instrument calibration or use standards prepared in same background composition of ambient air.

    -   The authors conjecture that major error sources arose from the mole fraction uncertainties of major components, e.g. $N_2$, $O_2$, Ar and $CO_2$, and uncertainty of pressure broadening coefficients. According to this opinion, the authors will add sentences at the end of discussion section as follow.
    -   "It is worth noting that the quality of the TPBC correction can be improved further by using quality standards with lower composition uncertainties, including $^{13}CO_2$ isotopologues and precisely measured broadening coefficients that are deduced from advanced line-shape functions such as Galatry and Rautian profiles."
    -   With regard to the isotopes ratio, please see the reply for general comment 2.

2.  A further comment is that the authors do not mention the isotopic composition of the $CO_2$ used to prepare their synthetic standards. While I assume all eight standards were prepared

with the same batch of $CO_2$ (and thus having the same $CO_2$ isotopic composition), this is worth mentioning (and handling) explicitly (preferably with the $\delta^{13}CCO_2$ of the pure $CO_2$ used). As CRDS is a single line spectroscopic technique, it is inherently isotopologue specific. Therefore, using a pure $CO_2$ source with a significantly different isotopic composition from the background atmosphere will induce a systematic bias in CRDS determinations of mixing ratio unless this effect is accounted for. The authors already cite Lee et al. (2006), which deals with this question (though for NDIR rather than CRDS (for which the problem is at its most extreme)), so I assume they are familiar with the issue.

- The authors understand this comment is very similar to first specific comment of RC1. The 12/13 ratio of $CO_2$ raw gas for gravimetric standards was similar to the atmospheric level approximately -11‰. The volumetric standards with prepared with the dry air and high purity $N_2$ (>99.999%). This suggests similar isotope ratios would occur across the prepared cylinders. For verification (calibration) of prepared gravimetric (volumetric) standards, the $CO_2$ mole fractions in them were verified by GC-FID, which measured total carbon isotopes. Therefore, the isotope effect were hardly discernable in this study. However, it might be the case that the isotope ratios of $CO_2$ in the "dry air" can vary or deviate from the $CO_2$ raw gas to cause some extent of discrepancy in the CRDS response. The authors will add sentences at the end of the section 2.1 as follow.

- The $^{12}C/^{13}C$ ratio of $CO_2$ raw gas for the gravimetric standards was similar to the atmospheric level of approximately -11‰, which suggests similar isotope ratios would occur across the prepared cylinders as determined by gravimetry and volumetry. Nevertheless, isotope effects biasing the CRDS response seemed to be hardly discernable in this study because verification (calibration) of the $CO_2$ mole fractions in the prepared gravimetric (volumetric) standards was carried out by GC-FID, which measured the total carbon isotopes."

**Specific Comments**
1. P1 L28, consider inserting 'all' between quantify and its, and remove "considerably"

- It will be corrected as suggested.

2. P3 L20, gases to become 'gas'

- It will be corrected as pointed out

---

## Author Comment (AC2) · 20 Jul 2017

The authors appreciate Dr. Kim's kind consideration of this manuscript. Please find our replies to the referee comments below.

General Comments

1.  I note that two tanks that had close-to-ambient ratios of N2/O2/Ar, namely

[Figure]

EB0006391 and ME0434, showed excellent agreement with values derived from CRDS prior to any correction (-0.01 and 0.09 $\mu$mol/mol, respectively, in Table 4), and the TPBC corrected values actually get worse. In addition, while the TPBC corrections overall seem to make a positive impact, the correction errors still remain quite larger than the 0.01% instrument precision error that the authors suggest should be the ultimate goal. Do the authors have any comments on what other error sources could remain that would explain these results (some of which seems to already be present at the end of the discussion section)?

- The CRDS employed in this study was calibrated against the gravimetric standard suite, the matrix compositions of which are very close to that of the atmosphere. Therefore, good agreements can be expected between the CRDS responses and the $CO_2$ mole fractions of EB0006391 and ME0434. Although, as pointed out, a worse agreement was found with the TPBC corrected values, this is within an acceptable margin considering the $CO_2$ mole fraction uncertainties of the employed cylinders, which are up to 0.1 % (Table 2). The authors conjecture that other error sources arose from imperfection in the regression analysis, mole fraction uncertainties of background gas compositions, uncertainties of pressure broadening coefficients, and instrumental drift. Accurate determination of the uncertainty budget requires further study, which is beyond the scope of this work. However, the authors will add the following sentence at the end of the discussion section.

- "It is worth noting that the quality of the TPBC correction can be improved further by using quality standards with lower composition uncertainties, including 13CO2 isotopologues and precisely measured broadening coefficients that are deduced from advanced line-shape functions such as Galatry and Rautian profiles." - With regard to the isotope ratio, please see the reply for specific comment 1.

2. Can the authors think of any scenarios outside of creating standard tanks from scratch that the TPBC correction would be necessary or beneficial?

- Dynamic mixing methods can be adapted to explore the nature of pressure broadening. The authors' impression with regard to improving the TPBC correction quality is that precise measurement of the corresponding absorption lines fitted by advanced line-shape functions such as Galatry and Rautian is needed.

Specific Comments

1. As the authors will know, the WS-CRDS technique measures only the main $12CO_2$ isotopologue. I think any effect from this can effectively be canceled out if all of the gas mixtures used in this study (listed in both Table 2 and 3) used $CO_2$ from the same source cylinder. I wonder if this is indeed the case, and whether the authors should briefly address this point somewhere in the manuscript.

- The volumetric standards were prepared with "dry air" and high-purity N2 (>99.999%). The 12/13 ratio of $CO_2$ raw gas for the gravimetric standards was similar to the atmospheric level of approximately -11‰.This suggests similar isotope ratios would occur across the prepared cylinders. For verification (calibration) of the prepared gravimetric (volumetric) standards, the $CO_2$ mole fractions in them were verified by GC-FID, which measured the total carbon isotopes. Therefore, the isotope effects were hardly discernable in this study. However, it might be the case that the isotope ratios of $CO_2$ in "dry air" can vary or deviate from those in the $CO_2$ raw gas to cause some extent of discrepancy in the CRDS response. The authors will add the following sentences at the end of section 2.1 as follow.

- "The 12C/13C ratio of $CO_2$ raw gas for the gravimetric standards was similar to the atmospheric level of approximately -11‰ which suggests similar isotope ratios would occur across the prepared cylinders as determined by gravimetry and volumetry. Nevertheless, isotope effects biasing the CRDS response seemed to be hardly discernable in this study because verification (calibration) of the $CO_2$ mole fractions in the prepared gravimetric (volumetric) standards was carried out by GC-FID, which measured the total carbon isotopes."

2. P3-L13: Was any correction to the concentrations applied based on the verification test on the GC, and if so how much? The authors state the verification test results were excellent (0.05 and 0.1 % $2\sigma$), but it would be interesting to see if those that looked worse in the verification test also showed larger deviation in the TPBC corrections. Perhaps this could be added in a supplementary section?

- Only "survivors" from the verification measurements for the gravimetric standards were used in this study. That is, outliers over the uncertainty of the verification measurement, identifying human error during the gas handling, were removed from the testing list. It should be noted that the weighing uncertainty is much less than that of the verification measurement. Additionally, the CO2 mole fraction uncertainty of the gravimetric mixtures included uncertainties associated with the weighing process, raw gas purities, and verification tests.

3. P3-L20: I think a more detailed description is needed for the static volumetric standard gas section. For example, line 22 mentions "dry air", is this some CO2-free zero air that was used as the "complementary gas" (using the terminology in ISO 6144), or does it just refer to what was already in the tank prior to the "high-purity N2" injection? Line 24 says the concentrations of the manometric cylinders were "confirmed" against the gravimetric standards: I would like clarification on whether the independent manometric values were confirmed by measurements against the gravimetric standards (on GC-FID?), and if so how the manometric vs gravimetric values compared, or if the values in the manometric tanks were "determined" from measurements against the gravimetric tanks. If the values were only confirmed, it would be nice to see how the values compared, perhaps in a supplementary section.

- The "dry air" referred to dehumidified air with CO2, which was already in the cylinder prior to the high-purity N2 injection. It was assumed that the high-purity N2 (> 99.999%) did not contain O2, Ar, and CO2 impurities; hence, it was possible to predict the mole fractions of the four components. Because of the daily variation of CO2, the CO2 mole fraction was given by the calibrated values against the gravimetric standards. The term

"manometric" was used to express the control of the mixing ratio using the volumetric ratio in this study; it will be toned down by replacing it with "volumetric mixing." The following sentences will be added in the corresponding section of the text.

- "Ambient air was collected with a pressurizing pump through a chemical moisture trap containing $Mg(ClO4)2$ in order to yield the complementary gas, namely dry air. The amount of $N2$ was then varied by diluting the dry air with high-purity $N2$ (> 99.999%), which eventually led to a variation in the mole fractions of the major components, $N2$, $O2$, $Ar$, and $CO2$. In this way, the mole fractions of the background gas composition can be easily predicted by using the measured pressure ratio of the filled gas. In the case of the $CO2$ mole fraction, three volumetric cylinders (EBXXXXXXX) were calibrated against the gravimetric standards (Table 2), because the mixing ratio of atmospheric $CO2$ varies each day. Eventually, the compositions of EB0006391 and ME0434 closely reflected the atmospheric ratio of the major components."

4. P4-L18: The numbers for the y-scale shown in Figure 4 (roughly -10 $\sim$ 5.5?) do not seem to match those in column 4 of Table 7 (-0.47 $\sim$ 0.60), but instead those in Table 4. Authors should check that this is only a graphing error and do not affect the conclusions of the paper. Tables 4 and 7: I understand the logic of the authors' choice of separating the two tables to match the flow of the manuscript, however I do find myself frequently comparing the N2-only vs TPBC corrected results. As such I would suggest that they be combined into one table, to represent an overview of the findings reported in this work, but I will leave that for the authors to decide.

- We apologize for the confusion. In Figure 7, DSTD-CRDS, as defined in P4-L34, denotes the deviation between the $CO2$ mole fraction of the standard and the corresponding CRDS response. However, in Table 7, the same value, DSTD-CRDS, was not given contrast to Table 4 (fifth column). As suggested, Table 4 and Table 7 will be combined to enhance the readability.

Technical Corrections
1. P1-L29: "not plausible" suggests that this can't be done in the future, which may be true, but we should still remain hopeful that substantial progress in the modeling front can still be made. Perhaps change to "not yet feasible" instead?

- This will be corrected as suggested.

2. P3-L20: I would suggest that the authors start a new paragraph for the section on the volumetrically prepared tanks.

- The preparation section will be separated and modified as suggested.

3. P3-L22: Is the "high-purity N2" used in the dilution different from the "ultra-high-purity nitrogen" mentioned in line 15? If they are the same, then I would advise using the same naming scheme for both.

- They are the same. "Ultra-high-purity nitrogen" will be replaced with "high-purity N2."

4. P3-L24: "comprised" -> "is comprised of"

- Dr. Kim might be referring to P3-L34 here. It will be corrected as suggested.

5. P3-L25: Perhaps mention which of the tanks reflect ratios close to ambient? I assume EB0006391 and ME0434?

- The following sentence will be added: "Eventually, the compositions of EB0006391 and ME0434 closely reflected the atmospheric ratio of N2, O2, Ar, and CO2."

6. P3-L40: "through a built-in diaphragm pump": Technically, I believe the pump pulls a vacuum after the cavity cell, whereas the authors' description gives the impression that air may go through the diaphragm pump into the cavity cell. Suggest editing this sentence to avoid ambiguity.

- Apologies for the ambiguity. The corresponding sentence will be corrected to "the optical cavity backed by a built-in diaphragm pump."

7. P3-L41: "inner" -> Did the authors mean "outer"?

- This will be revised as suggested.

8. P4-L8: "gravimetric standards" -> add "described in Table 3" after. How were these standards prepared in terms of N2, O2, and Ar? I assume at ambient ratios? This may be an important point, as the authors use the calibrations from these tanks as "truth".

- The corresponding sentence will be corrected to "gravimetric standards, in which the N2, O2, and Ar ratio is close to that in the atmosphere ratio, with CO2 concentrations..."

9. p5-L13: Include reference for "HITRAN2004"?

- The reference was included in the references section.

10. P5-L16: "that" -> "those"

- This will be corrected as suggested.

11. Table 6: I do not follow the author's foot note "1 and 2 denote values obtained in each study" for this table. I assume the numbers in this table were derived using the PBC's in Table 5 with the known N2, O2, and Ar ratios? But, aren't the HITRAN numbers calculated the same way, or am I mistaken? The footnote almost seems more appropriate for Table 5, where the PBC values in the table were taken from each study, but then are the HITRAN numbers different in this regard? Please clarify.

- Thank you for the comment. The footnote will be deleted. To enhance readability, the following sentence will be added as a footnote.

- "Pressure broadenings were estimated without Ar due to the absence of a broadening coefficient in the corresponding studies."

Please also note the supplement to this comment:
https://www.atmos-meas-tech-discuss.net/amt-2017-54/amt-2017-54-AC2-supplement.pdf

---

## Author Response (AR1)

**Reply to RC1**

Jeongsoon Lee (Corresponding author)

leejs@kriss.re.kr

The authors appreciate Dr. Kim's kind consideration of this manuscript. Please find our replies to the referee comments below.

**General Comments**

1. I note that two tanks that had close-to-ambient ratios of $N_2/O_2/Ar$, namely EB0006391 and ME0434, showed excellent agreement with values derived from CRDS prior to any correction (-0.01 and 0.09 µmol/mol, respectively, in Table 4), and the TPBC corrected values actually get worse. In addition, while the TPBC corrections overall seem to make a positive impact, the correction errors still remain quite larger than the 0.01% instrument precision error that the authors suggest should be the ultimate goal. Do the authors have any comments on what other error sources could remain that would explain these results (some of which seems to already be present at the end of the discussion section)?

   - The CRDS employed in this study was calibrated against the gravimetric standard suite, the matrix compositions of which are very close to that of the atmosphere. Therefore, good agreements can be expected between the CRDS responses and the $CO_2$ mole fractions of EB0006391 and ME0434. Although, as pointed out, a worse agreement was found with the TPBC corrected values, this is within an acceptable margin considering the $CO_2$ mole fraction uncertainties of the employed cylinders, which are up to 0.1 % (Table 2). The authors conjecture that other error sources arose from imperfection in the regression analysis, mole fraction uncertainties of background gas compositions, uncertainties of pressure broadening coefficients, and instrumental drift. Accurate determination of the uncertainty budget requires further study, which is beyond the scope of this work. However, the authors will add the following sentence at the end of the discussion section.
   - "It is worth noting that the quality of the TPBC correction can be improved further by using quality standards with lower composition uncertainties, including $^{13}CO_2$ isotopologues and precisely measured broadening coefficients that are deduced from advanced line-shape functions such as Galatry and Rautian profiles."
   - With regard to the isotope ratio, please see the reply for specific comment 1.

2. Can the authors think of any scenarios outside of creating standard tanks from scratch that the TPBC correction would be necessary or beneficial?

- Dynamic mixing methods can be adapted to explore the nature of pressure broadening. The authors' impression with regard to improving the TPBC correction quality is that precise measurement of the corresponding absorption lines fitted by advanced line-shape functions such as Galatry and Rautian is needed.

**Specific Comments**

1. As the authors will know, the WS-CRDS technique measures only the main $^{12}CO_2$ isotopologue. I think any effect from this can effectively be canceled out if all of the gas mixtures used in this study (listed in both Table 2 and 3) used $CO_2$ from the same source cylinder. I wonder if this is indeed the case, and whether the authors should briefly address this point somewhere in the manuscript.

- The volumetric standards were prepared with "dry air" and high-purity $N_2$ (>99.999%). The 12/13 ratio of $CO_2$ raw gas for the gravimetric standards was similar to the atmospheric level of approximately -11‰. This suggests similar isotope ratios would occur across the prepared cylinders. For verification (calibration) of the prepared gravimetric (volumetric) standards, the $CO_2$ mole fractions in them were verified by GC-FID, which measured the total carbon isotopes. Therefore, the isotope effects were hardly discernable in this study. However, it might be the case that the isotope ratios of $CO_2$ in "dry air" can vary or deviate from those in the $CO_2$ raw gas to cause some extent of discrepancy in the CRDS response. The authors will add the following sentences at the end of section 2.1 as follow.
- "The $^{12}C/^{13}C$ ratio of $CO_2$ raw gas for the gravimetric standards was similar to the atmospheric level of approximately -11‰, which suggests similar isotope ratios would occur across the prepared cylinders as determined by gravimetry and volumetry. Nevertheless, isotope effects biasing the CRDS response seemed to be hardly discernable in this study because verification (calibration) of the $CO_2$ mole fractions in the prepared gravimetric (volumetric) standards was carried out by GC-FID, which measured the total carbon isotopes."

2. P3-L13: Was any correction to the concentrations applied based on the verification test on the GC, and if so how much? The authors state the verification test results were excellent (0.05 and 0.1 % 2σ), but it would be interesting to see if those that looked worse in the verification test also showed larger deviation in the TPBC corrections. Perhaps this could be added in a supplementary section?

- Only "survivors" from the verification measurements for the gravimetric standards were used in this study. That is, outliers over the uncertainty of the verification measurement, identifying human error during the gas handling, were removed from the testing list. It should be noted that the weighing uncertainty is much less than that of the verification measurement. Additionally, the $CO_2$ mole fraction uncertainty of the gravimetric mixtures included

uncertainties associated with the weighing process, raw gas purities, and verification tests.

3. P3-L20: I think a more detailed description is needed for the static volumetric standard gas section. For example, line 22 mentions "dry air", is this some $CO_2$-free zero air that was used as the "complementary gas" (using the terminology in ISO 6144), or does it just refer to what was already in the tank prior to the "high-purity $N_2$" injection? Line 24 says the concentrations of the manometric cylinders were "confirmed" against the gravimetric standards: I would like clarification on whether the independent manometric values were confirmed by measurements against the gravimetric standards (on GC-FID?), and if so how the manometric vs gravimetric values compared, or if the values in the manometric tanks were "determined" from measurements against the gravimetric tanks. If the values were only confirmed, it would be nice to see how the values compared, perhaps in a supplementary section.

- The "dry air" referred to dehumidified air with $CO_2$, which was already in the cylinder prior to the high-purity $N_2$ injection. It was assumed that the high-purity $N_2$ (> 99.999%) did not contain $O_2$, Ar, and $CO_2$ impurities; hence, it was possible to predict the mole fractions of the four components. Because of the daily variation of $CO_2$, the $CO_2$ mole fraction was given by the calibrated values against the gravimetric standards. The term "manometric" was used to express the control of the mixing ratio using the volumetric ratio in this study; it will be toned down by replacing it with "volumetric mixing." The following sentences will be added in the corresponding section of the text.
- "Ambient air was collected with a pressurizing pump through a chemical moisture trap containing $Mg(ClO_4)_2$ in order to yield the complementary gas, namely dry air. The amount of $N_2$ was then varied by diluting the dry air with high-purity $N_2$ (> 99.999%), which eventually led to a variation in the mole fractions of the major components, $N_2$, $O_2$, Ar, and $CO_2$. In this way, the mole fractions of the background gas composition can be easily predicted by using the measured pressure ratio of the filled gas. In the case of the $CO_2$ mole fraction, three volumetric cylinders (EBXXXXXXX) were calibrated against the gravimetric standards (Table 2), because the mixing ratio of atmospheric $CO_2$ varies each day. Eventually, the compositions of EB0006391 and ME0434 closely reflected the atmospheric ratio of the major components."

4. P4-L18: The numbers for the y-scale shown in Figure 4 (roughly -10 ~ 5.5?) do not seem to match those in column 4 of Table 7 (-0.47 ~ 0.60), but instead those in Table 4. Authors should check that this is only a graphing error and do not affect the conclusions of the paper. Tables 4 and 7: I understand the logic of the authors' choice of separating the two tables to match the flow of the manuscript, however I do find myself frequently comparing the $N_2$-only vs TPBC corrected results. As such I would suggest that they be combined into one table, to represent an overview of the findings reported in this work, but I will leave that for the authors to decide.

- We apologize for the confusion. In Figure 7, $D_{STD\text{-}CRDS}$, as defined in P4-L34, denotes the deviation between the $CO_2$ mole fraction of the standard and the corresponding CRDS response. However, in Table 7, the same value, $D_{STD\text{-}CRDS}$, was not given contrast to Table 4 (fifth column). As suggested, Table 4 and Table 7 will be combined to enhance the readability.

**Technical Corrections**

1. P1-L29: "not plausible" suggests that this can't be done in the future, which may be true, but we should still remain hopeful that substantial progress in the modeling front can still be made. Perhaps change to "not yet feasible" instead?

- This will be corrected as suggested.

2. P3-L20: I would suggest that the authors start a new paragraph for the section on the volumetrically prepared tanks.

- The preparation section will be separated and modified as suggested.

3. P3-L22: Is the "high-purity $N_2$" used in the dilution different from the "ultra-high-purity nitrogen" mentioned in line 15? If they are the same, then I would advise using the same naming scheme for both.

- They are the same. "Ultra-high-purity nitrogen" will be replaced with "high-purity $N_2$."

4. P3-L24: "comprised" -> "is comprised of"

- Dr. Kim might be referring to P3-L34 here. It will be corrected as suggested.

5. P3-L25: Perhaps mention which of the tanks reflect ratios close to ambient? I assume EB0006391 and ME0434?

- The following sentence will be added: "Eventually, the compositions of EB0006391 and ME0434 closely reflected the atmospheric ratio of $N_2$, $O_2$, Ar, and $CO_2$."

6. P3-L40: "through a built-in diaphragm pump": Technically, I believe the pump pulls a vacuum after the cavity cell, whereas the authors' description gives the impression that air may go through the diaphragm pump into the cavity cell. Suggest editing this sentence to avoid ambiguity.

- Apologies for the ambiguity. The corresponding sentence will be corrected to "the optical cavity

backed by a built-in diaphragm pump."

7. P3-L41: "inner" -> Did the authors mean "outer"?

- This will be revised as suggested.

8. P4-L8: "gravimetric standards" -> add "described in Table 3" after. How were these standards prepared in terms of N2, O2, and Ar? I assume at ambient ratios? This may be an important point, as the authors use the calibrations from these tanks as "truth".

- The corresponding sentence will be corrected to "gravimetric standards, in which the $N_2$, $O_2$, and Ar ratio is close to that in the atmosphere ratio, with $CO_2$ concentrations..."

9. p5-L13: Include reference for "HITRAN2004"?

- The reference was included in the references section.

10. P5-L16: "that" -> "those"

- This will be corrected as suggested.

11. Table 6: I do not follow the author's foot note "1 and 2 denote values obtained in each study" for this table. I assume the numbers in this table were derived using the PBC's in Table 5 with the known N2, O2, and Ar ratios? But, aren't the HITRAN numbers calculated the same way, or am I mistaken? The footnote almost seems more appropriate for Table 5, where the PBC values in the table were taken from each study, but then are the HITRAN numbers different in this regard? Please clarify.

- Thank you for the comment. The footnote will be deleted. To enhance readability, the following sentence will be added as a footnote.
- "Pressure broadenings were estimated without Ar due to the absence of a broadening coefficient in the corresponding studies."

**Reply to RC2**

Jeongsoon Lee (Corresponding author)

leejs@kriss.re.kr

The authors appreciate Dr. Loh's kind consideration of this manuscript. Please find our replies to the referee comments below.

**General Comments**

1.  The authors present a set of total pressure broadening coefficients (TPBCs) that substantially improve agreement between CRDS determined $CO_2$ mixing ratios and the mixing ratios assigned to each tank during gravimetric or manometric preparation. However, the use of TPBCs does not reduce the discrepancy to within the World Meteorological Organization's $CO_2$ inter-laboratory compatibility goal of +/- 0.1 umol/mol (in the Northern Hemisphere, and 0.05 umol/mol in the Southern Hemisphere). As such, I would urge the authors to consider appending something similar to the following to the end of their abstract.

    P1, L20: "… instrument calibration, or better still, use standards prepared with ambient air."

    Additionally, I would like the authors to consider adding a sentence or two to this effect in their discussion section.

    -   Thank you for the suggestion. Authors will add sentence as follow.
    -   P1, L20: "…. Instrument calibration or use standards prepared in same background composition of ambient air.

    -   The authors conjecture that major error sources arose from the mole fraction uncertainties of major components, e.g. $N_2$, $O_2$, Ar and $CO_2$, and uncertainty of pressure broadening coefficients. According to this opinion, the authors will add sentences at the end of discussion section as follow.
    -   "It is worth noting that the quality of the TPBC correction can be improved further by using quality standards with lower composition uncertainties, including $^{13}CO_2$ isotopologues and precisely measured broadening coefficients that are deduced from advanced line-shape functions such as Galatry and Rautian profiles."
    -   With regard to the isotopes ratio, please see the reply for general comment 2.

2.  A further comment is that the authors do not mention the isotopic composition of the $CO_2$ used to prepare their synthetic standards. While I assume all eight standards were prepared

with the same batch of $CO_2$ (and thus having the same $CO_2$ isotopic composition), this is worth mentioning (and handling) explicitly (preferably with the $\delta^{13}CCO_2$ of the pure $CO_2$ used). As CRDS is a single line spectroscopic technique, it is inherently isotopologue specific. Therefore, using a pure $CO_2$ source with a significantly different isotopic composition from the background atmosphere will induce a systematic bias in CRDS determinations of mixing ratio unless this effect is accounted for. The authors already cite Lee et al. (2006), which deals with this question (though for NDIR rather than CRDS (for which the problem is at its most extreme)), so I assume they are familiar with the issue.

- The authors understand this comment is very similar to first specific comment of RC1. The 12/13 ratio of $CO_2$ raw gas for gravimetric standards was similar to the atmospheric level approximately -11‰. The volumetric standards with prepared with the dry air and high purity $N_2$ (>99.999%). This suggests similar isotope ratios would occur across the prepared cylinders. For verification (calibration) of prepared gravimetric (volumetric) standards, the $CO_2$ mole fractions in them were verified by GC-FID, which measured total carbon isotopes. Therefore, the isotope effect were hardly discernable in this study. However, it might be the case that the isotope ratios of $CO_2$ in the "dry air" can vary or deviate from the $CO_2$ raw gas to cause some extent of discrepancy in the CRDS response. The authors will add sentences at the end of the section 2.1 as follow.

- The $^{12}C/^{13}C$ ratio of $CO_2$ raw gas for the gravimetric standards was similar to the atmospheric level of approximately -11‰, which suggests similar isotope ratios would occur across the prepared cylinders as determined by gravimetry and volumetry. Nevertheless, isotope effects biasing the CRDS response seemed to be hardly discernable in this study because verification (calibration) of the $CO_2$ mole fractions in the prepared gravimetric (volumetric) standards was carried out by GC-FID, which measured the total carbon isotopes."

**Specific Comments**

5. P1 L28, consider inserting 'all' between quantify and its, and remove "considerably"

- It will be corrected as suggested.

6. P3 L20, gases to become 'gas'

- It will be corrected as pointed out

**Validation of spectroscopic gas analyzer accuracy using gravimetric standard gas mixtures: Impact of background gas composition on $CO_2$ quantitation by cavity ring-down spectroscopy**

Jeong Sik Lim, Miyeon Park, Jinbok Lee, Jeongsoon Lee

Center for Gas Analysis, Metrology for Quality of Life, Korea Research Institute of Standards and Science (KRISS), Gajeong-ro 267, Yuseong-gu, Daejeon 34113, Republic of Korea

*Correspondence to*: Jeongsoon Lee (leejs@kriss.re.kr)

**Abstract.** Effect of background gas composition on the measurement of $CO_2$ levels was investigated by wavelength-scanned cavity ring-down spectrometry (WS-CRDS) employing a spectral line centered at the R(1) of the $(3\ 0^0\ 1)_{III} \leftarrow (0\ 0\ 0)$ band. For this purpose, eight cylinders with various gas compositions were gravimetrically and volumetrically prepared within $2\sigma = 0.1$ %, and these gas mixtures were introduced into the WS-CRDS analyzer calibrated against standards of ambient air composition. Depending on the gas composition, deviations between CRDS-determined and gravimetrically (or volumetrically) assigned $CO_2$ concentrations ranged from -9.77 to 5.36 μmol/mol, e.g., excess $N_2$ exhibited a negative deviation, whereas excess Ar showed a positive one. The total pressure broadening coefficients (TBPCs) obtained from the composition of $N_2$, $O_2$ and Ar thoroughly corrected the deviations up to -0.5–0.6 μmol/mol, while these values were -0.43–1.43 μmol/mol considering PBCs induced by only $N_2$. The use of TBPCs enhanced deviations to be corrected to ~0.15 %. Furthermore, the above correction linearly shifted CRDS responses for a wide extent of TPBCs ranging from 0.065 to 0.081 $cm^{-1}$ $atm^{-1}$. Thus, accurate measurements using optical intensity-based techniques such as WS-CRDS require TBPC-based instrument calibration or use standards prepared in same background composition of ambient air.

Copyright statement: The authors warrant that the article is original, is not under consideration by another journal, and has not been previously published.

[revised manuscript text omitted]